# The Effects of 12-Week Prebiotic Supplementation on General Wellness and Exercise-Induced Gastrointestinal Symptoms in Recreationally Trained Endurance Athletes: A Triple-Blind Randomised Controlled Pilot Trial

**DOI:** 10.3390/nu17213390

**Published:** 2025-10-28

**Authors:** Lewis A. Gough, Anthony Weldon, Cain C. T. Clark, Anthony Young, Charlie J. Roberts, Neil D. Clarke, Meghan A. Brown, Rachel Williams

**Affiliations:** 1Research for Human Performance and Health Laboratory, Birmingham City University, Birmingham B42 2LR, UK; lewis.gough@bcu.ac.uk (L.A.G.); anthony.weldon@bcu.ac.uk (A.W.); anthony.young@mail.bcu.ac.uk (A.Y.); charlie.roberts@bcu.ac.uk (C.J.R.); rachel.williams2@bcu.ac.uk (R.W.); 2Carnegie School of Sport, Leeds Beckett University, Leeds LS1 3HE, UK; meghan.brown@leedsbeckett.ac.uk

**Keywords:** prebiotics, galactooligosaccharides (GOS), endurance athletes, gastrointestinal symptoms, wellbeing

## Abstract

**Background/Objectives**: Ingestion of galactooligosaccharides (GOSs) or GOS mixtures has been purported to improve exercise-induced gastrointestinal (GI) distress and post-exercise recovery. However, the effects have not been explored in recreationally trained endurance athletes. This triple-blind randomised controlled trial, therefore, investigated whether 12 weeks of B-GOS^®^ supplementation affects gastrointestinal comfort and psychological wellbeing in recreational athletes. **Methods**: Eighteen physically active individuals (12 males, 8 females, 44 ± 14 years, 1.7 ± 0.1 m and 73 ± 14 kg) volunteered for this study. Participants were assigned to independent groups in a placebo-controlled, triple-blind manner via stratified randomisation. A 20 min run at 80% VO_2max_ was completed, with measures for GI distress and Competitive State Anxiety Inventory-2 questionnaire (CSAI-2) pre- and post-exercise. A 12-week supplementation period then ensued, where participants ingested either 3.65 g of B-GOS or an appearance-matched maltodextrin placebo. During this time, physical activity levels (IPAQ-7), general stress (REST-Q), mental wellbeing (WEMWBS), and sleep (core consensus sleep diary) were measured at regular time points. **Results**: There were no significant differences in VO_2max_ (*p* = 0.437), GI discomfort (*p* = 0.227), or CSAI-2 (*p* = 0.739–0.954) from pre- to post-exercise at any time point or between conditions. Over the 12 weeks there were no significant differences between B-GOS and placebo in IPAQ-7 (*p* = 0.144–0.723), REST-Q (*p* = 0.282–0.954), WEMWBS (B-GOS pre = 51 ± 10, post = 53 ± 7; PLA pre = 51 ± 4, post 54; *p* = 0.862), or sleep (*p* = 0.065–0.992). The linear mixed model suggests that some may benefit on an individual level in terms of WEMWBS, general stress score, recovery-related scores, sleep, and sport-specific recovery score. **Conclusions**: There were no group benefits of B-GOS supplementation compared with placebo, although the individual variation may warrant further research in larger sample sizes and longer-duration studies.

## 1. Introduction

It is widely reported that a reduction in beneficial members of the genus Bifidobacterium occurs with age, and this may compromise immunity, mental wellbeing, and sleep [1]. Prebiotics, especially in the form of transgalactooligosaccharides, are substrates that could have a positive influence on aspects of health [2] by increasing Bifidobacterium levels. A prebiotic galactooligosaccharide (GOS) composition derived from lactose (milk), B-GOS (Bimuno^®^), has garnered scientific attention due to its ability to increase Bifidobacterium levels in humans [3]. Reasons why this may be beneficial are predominantly related to modulation of the microbiota–gut–brain axis and reduction in chronic system inflammation, although other mechanisms are purported (for a review, see [2]). Indeed, Vulevic et al. [3] reported, in forty elderly volunteers (25 women and 15 men, average age: 70 years), that bacteroides and bifidobacteria were significantly increased following 5.5 g/day of B-GOS for 10 weeks. It is intuitive to suggest that changes in Bifidobacterium levels may be worthwhile in athletes to induce positive change in the microbiota–gut–brain axis and reduce chronic system inflammation, as reports of compromised immunity, reduced mental wellbeing, and disturbed sleep are prevalent [2].

In two key studies investigating rugby [4] and non-elite adult participants [5], positive effects of B-GOS were specifically reported across 24-week and 3-week periods, respectively. Generally, reductions in the length of URTIs by approximately 2 days were reported during 24 weeks of B-GOS supplementation within a group of 33 elite rugby players [4]. This is beneficial given that this population is under higher training stress and URTIs are a common problem. In a further study, reductions in resting cortisol upon awakening and improved emotional processing were reported following B-GOS after only 3 weeks of B-GOS supplementation [5]. A reduction in gastrointestinal (GI) side effects have also been reported after 4 weeks supplementation of a blended compound (probiotic formulation comprising of lactobacillus acidophilus and bifidobacterium longum) [6]. Comparisons across studies are difficult due to the inconsistencies in methodology (e.g., time frame of supplementation, blended ingredients, and varied measurement scales); however, they collectively suggest that ingestion of B-GOS or mixtures containing B-GOS can improve various aspects of GI symptoms and potentially mood state. These effects may lead to improved training availability (through reduced length of URTIs) or quality (due to fewer GI symptoms and improved mood) and therefore greater competitive success and/or general health.

The administration of B-GOS in recreationally trained endurance athletes is limited in the literature. To date, B-GOS has been administered to healthy individuals or those with GI conditions such as irritable bowel syndrome (IBS) or Ulcerative Colitis [7,8], or elite athletes [4]. The administration in recreationally trained endurance athletes is important given that this population is susceptible to URTIs and GI discomfort [9] and may suffer from mood disturbance due to the high levels of training volume [10], alongside employment and life stressors. Given that previous studies, albeit in different populations, have reported benefits in URTI length, GI discomfort, and mood, it is sensible to suggest that the ingestion of B-GOS could provide important benefits to endurance athletes. Furthermore, since such products are also marketed to recreational athletes and are commercially available, it is important to ascertain any potential benefits to justify their usage within this specific population given they may elicit different responses compared with elite athletes or those with underlying health conditions. The aim of this pilot study was, therefore, to investigate the effects of 12 weeks of B-GOS supplementation on general wellness and exercise-induced GI symptoms in recreationally trained endurance athletes.

## 2. Materials and Methods

### 2.1. Participants

Eighteen physically active individuals (12 = males, 6 = females, 44 ± 14 years, 1.7 ± 1.2 m and 73 ± 14 kg; VO_2max_ = 43.9 ± 6.7 mL·kg^−1^ BM·min^−1^) who regularly participated in endurance exercise were recruited for this study through a combination of purposive and snowball sampling. Participants were all over the age of 18 and participated in running training twice per week. It was verbally confirmed they were free of injury and had not ingested antibiotics for at least 3 months prior to the study. They were also free of infection upon study enrolment. Finally, participants had stable dietary intake and were asked to keep this the same throughout the study. Specifically, participants were asked to not attempt any new dietary strategy or any new supplements alongside B-GOS.

Participants were assigned to either the B-GOS or placebo group (*n* = 9 per group) in a triple-blind manner via stratified randomisation. The randomisation was conducted by a member of staff outside of the study, and the statistics were carried out by a member of the research team with only the descriptors “treatment A” and “treatment B”. Supplements were also provided in concealed sachets labelled in the same way as the analysis and were identical in their appearance and taste. The treatments (B-GOS or PLA) were only decoded once analysis was completed.

Prior to any experimental procedures, participants provided verbal and written consent and completed a Physical Activity Readiness Questionnaire (PAR-Q). The study received institutional ethical approval (Williams/BCU/11406). A priori power calculation was not conducted due to the resource constraints (budget and high participant burden) of the study and the fact it was a pilot study where hypothesis testing was not the primary focus [11].

### 2.2. Experimental Overview

Participants visited the laboratory on three separate occasions in a triple-blind independent group study. Participants arrived at the laboratory, which was regulated to 21 °C (±1 °C), at a similar time of day (+two hours) to account for the effects of circadian rhythms [12]. Participants were advised to maintain their usual nutritional practice throughout the study and to avoid any strenuous physical activity 24 h before each laboratory visit. For the initial laboratory visit, participants completed an incremental running test to voluntary exhaustion to determine their maximum rate of oxygen uptake (VO_2max_). For the second and third visits, participants returned to the laboratory to perform a 20 min run at a constant speed that was equivalent to 80% of their VO_2max_ on a motorised treadmill with a 1% incline [13]. Between the second and third laboratory visits, participants took part in a 12-week supplementation period whilst being assigned to one of two independent groups (B-GOS or Placebo) in a triple-blind manner via stratified randomisation. During the 12-week supplementation period, participants were given a logbook containing a daily sleep diary, a weekly physical activity questionnaire, a bi-weekly mental health and wellbeing questionnaire, and a recovery–stress questionnaire to complete every 4 weeks (Figure 1). Either the Bimuno^®^ (Clasado Biosciences, Reading, UK) (3.65 g/day Bimuno galactooligosaccharides, equivalent to approximately 2.75 g active GOS) or Placebo (3.65 g/day maltodextrin) product was consumed by the subjects for 12 weeks. The supplements were administered as a powder sachet to be mixed in fluid. A schematic representation of the study design is provided in Figure 1.

### 2.3. Maximum Rate of Oxygen Uptake Test and Familiarisation

Anthropometric measures for body mass (813, SECA, Hamburg, Germany) and height (Stadiometer, Holtain Ltd., Crosswell, UK) were taken, along with resting heart rate (Polar H10 Sensor, Polar Electro, Kempele, Finland) after a 5 min seated rest. Resting blood lactate (Lactate Pro 2, Arkay, Kyoto, Japan) measures via capillary finger-prick sampling (5 µL) were collected prior to the incremental running test on a motorised treadmill (Woodway Pro, USA Inc., Waukesha, WI, USA). Participants completed an individualised warm-up on the treadmill based upon knowledge of partaking in endurance exercise. In the first visit, the maximal rate of oxygen uptake was measured using a breath-by-breath respiratory device (Metalyzer 3B, Cortex Biophysik GmbH, Leipzig, Germany) that was calibrated as per the manufacturer’s instructions. The running speed for the incremental running test was set to 7.0 km/h for women and 9.0 km/h for men, with an increase of 1.0 km/h every 4 min [14]. The participant’s heart rate (Polar H10 Sensor, Polar Electro, Kempele, Finland), whole-body Rate of Perceived Exertion (RPE) [15], and blood lactate measures were collected during the incremental running test after every 4 min [16]. Once blood lactate measures reached 4.0 mmol/L, participants decided whether they wanted the running speed or incline to be increased by 1.0 km/h or 1% every minute until the participant reached volitional exhaustion [16]. The end of the test was considered when the participant was unable to continue to run on the treadmill. The VO_2max_ was confirmed following the recommendation of Bird and Davison [17], where participants achieved 10 b·min^−1^ of age predicted maximal HR (220 b·min^−1^—age), a respiratory exchange ratio (RER) of ≥1.15, an RPE between 19 and 20 at the end of exercise, and a blood lactate level of >8 mmol/L.

Following a 30 min recovery, participants were then familiarised to the various questionnaires and scales in the experiment. In particular, the Modified Visual Analogue Scale (mVAS) was completed in relation to the GI distress experiences during the test. They also completed the Competitive State Anxiety Inventory-2 questionnaire [18].

### 2.4. Experimental Procedures

The next two visits consisted of visiting the laboratory to complete a 20 min run at constant speed (80% VO_2max_) on a motorised treadmill with a 1% incline under temperate conditions (21 °C ± 1 °C). This was utilised as it is known to negatively impact gut permeability and elevate gut damage markers [13]. This was completed in week 0 and week 12 (pre- and post-supplementation period).

Participants completed the CSAI-2 questionnaire prior to the experimental protocol [18]. Resting heart rate was measured, and a baseline blood lactate sample was collected (5 µL). An individualised warm-up then took place prior to the exercise protocol, and this was based upon the participant’s knowledge of participating in endurance exercise (this was replicated each time). The 20 min test then began, and breath-by-breath analysis was used to determine if participants were exercising at 80% of their VO_2max_ from the pre-experimental VO_2max_ test (Metalyzer 3B, Cortex Biophysik GmbH, Leipzig, Germany), with small adjustments in speed if required. The participant’s heart rate (Polar H10 Sensor, Polar Electro, Finland) and whole-body RPE [15] were recorded every 1 min for the duration of the 20 min exercise protocol, with a capillary blood sample for lactate being taken immediately after exercise (results not reported). After the exercise protocol and in the same visit, participants completed the mVAS, which was recorded as a subjective measure in relation to GI distress which consisted of three areas: upper, lower, and other GI symptoms [19].

Upon leaving the laboratory, participants were then provided with a logbook containing diaries, questionnaires, and scales as used in previous research: self-reported 4-day dietary diary [20] (data not reported, used only to encourage compliance), short form International Physical Activity Questionnaire (IPAQ-7) [21], Core Consensus Sleep Dairy [22], Warwick-Edinburgh Mental Wellbeing Scale (WEMWBS) [23], mVAS for GI distress (mVAS) [19], and recovery–stress questionnaire (RESTQ) [24]. These were requested from participants at various time points during the 12-week supplementation period (Figure 1). The timing of questionnaires was decided by the recommended time frame within each respective questionnaire and by balancing this with the participant burden. Prior to commencing the study, a small number of endurance recreational runners (*n* = 5) reviewed the planned methods and provided suggestions on timings. Logbooks were completed using the conventional pen-and-paper method and were returned to the lead researcher during the final laboratory visit. Participants were given a 12-week supply of the Bimuno^®^ or placebo supplement to take daily (3.65 g). The placebo was maltodextrin (white powder), identical in appearance to the Bimuno^®^ supplement. Participants returned the empty sachets in week 12 as a method to encourage compliance. On the third visit to the laboratory, the methodology described above was repeated (CSAI-2, 20 min run, heart rate, RPE, blood lactate, and mVAS). Data collected from the logbooks were inputted into a Microsoft encrypted database (Excel 365, Microsoft, Mountain View, CA, USA).

### 2.5. Statistical Analysis

We utilised linear mixed modelling (LMM) to examine the effects of group (B-GOS vs. Placebo), time (pre vs. post), and their interaction on all outcome variables, with individual participants included as a random intercept to account for repeated measures. Random slopes for time were initially considered but were then omitted, as they did not improve model fit and occasionally produced singular fits. Missing data were handled via listwise deletion. Model parameters were estimated using maximum likelihood, whilst Satterthwaite’s method was used to approximate degrees of freedom and conduct significance testing. Model fit was assessed using Akaike Information Criterion (AIC), Bayesian Information Criterion (BIC), and Likelihood Ratio Tests, while variance explained by fixed and random effects was evaluated using Nakagawa’s R^2^, where both marginal R^2^ (variance explained by fixed effects only) and conditional R^2^ (variance explained by both fixed and random effects) were calculated and reported. Models were evaluated for appropriateness, and in instances where the LMM was not appropriate, due to issues such as singular fits or zero random effect variance, a linear model (LM) was used instead.

In instances where the overall model indicated a statistically significant effect, post-hoc pairwise comparisons were conducted using estimated marginal means with Tukey’s adjustment for multiple comparisons, which controls the family-wise error rate. Additionally, Cohen’s d effect sizes were computed for each contrast by using the residual standard deviation of the model and expressed as trivial (<0.2), small (0.2–0.6), moderate (0.6–1.2), large (1.2–2.0), and very large (≥2.0) [25].

Statistical significance was accepted, a priori, at *p* < 0.05, and 95% confidence intervals (CIs) were reported where applicable. Model assumptions, including normality of residuals and homoscedasticity, were visually inspected using diagnostic plots. All statistical analyses were conducted using R (R Core Team (2018); R: A Language and environment for statistical computing [computer software; retrieved from https://cran.r-project.org/ (accessed on 22 August 2025), with the lme4, lmerTest, performance, and emmeans packages]).

## 3. Results

### 3.1. Summary and Adherence

Descriptive statistics for all primary and secondary outcomes are presented in Table 1. Briefly (a detailed overview is presented in the Appendix A), baseline values were comparable between the B-GOS and placebo groups across all measures (Cognitive A-state: B-GOS = 45.2 ± 8.3, Placebo = 44.9 ± 7.9; WEMWBS: B-GOS = 52.1 ± 6.8, Placebo = 51.8 ± 6.5). Physical activity, assessed via the IPAQ, did not significantly differ between groups or over time. Adherence was confirmed by the return of empty sachets from all participants, which all participants completed, and no adverse events were reported in either group. 

### 3.2. Group, Time, and Group × Time Effects

Linear mixed modelling indicated that no main effects of group or time nor interaction effects of group x time were evident for Somatic A State, state self-confidence, overall gut discomfort, upper GIS, lower GIS, other GIS, q3Sleep, q4Sleep, q8Sleep, q9Sleep, q10Sleep, q12aSleep, q13aSleep, q1IPAQ, q2IPAQ, q3IPAQ, q4IPAQ, q5IPAQ, q6IPAQ, q7IPAQ, WEMWBS, general stress, recovery-related, or sport-specific recovery scores. Although these effects were not significant, corresponding effect sizes (Cohen’s d) and 95% confidence intervals are reported in Table 1 to evidence the magnitude and precision of the observed estimates. Most non-significant effects were small (D < 0.3), indicating minimal practical differences between conditions.

For Cognitive A state, we noted a significant main effect of time (β = −17.56, *p* < 0.001, 95% CI: −20.37, −14.75), where post-hoc pairwise comparisons found that post-test scores were significantly lower than pre-test ones (mean difference: 17.5, *p* < 0.001, D: 4.5). However, LMM indicated no significant main effects of group or interaction effects.

For sport-specific score, we found a significant main effect for group (β = 1.99, *p* = 0.026, 95% CI: 0.25, 3.72). Post-hoc comparison showed that Group A scores were significantly lower than Group B scores (mean difference: 1.95, *p* = 0.005, D: 1.15). However, LMM indicated no significant main effects of time or interaction effects.

### 3.3. Explanatory Power of Individual Differences

The additional Explanatory Power of Random Effects (i.e., individual participants) was evaluated using Nakagawa’s R^2^, reported as marginal R^2^ (fixed effects only) and conditional R^2^ (fixed + random effects) (Table 1). For twelve variables, inclusion of random effects notably increased explained variance—by over 50% in several cases (e.g., q3Sleep = 0.13 to 0.68; WEMWBS = 0.03 to 0.76; recovery-related score = 0.02 to 0.67), highlighting the influence of individual differences on these measures.

## 4. Discussion

This study investigated the effects of 12 weeks of B-GOS supplementation on general wellness and exercise-induced GI symptoms in recreationally trained endurance athletes. In accord with this aim, we found that B-GOS supplementation yielded no statistically significant effects across the variables measures in this study. There were no group × time interaction effects for all variables and no main effects for time bar Cognitive A state. Given that there were no adverse events reported, we can conclude from this pilot that the ingestion of B-GOS was tolerated and feasible on the basis that we experienced suitable adherence, tolerability, and no dropouts throughout the study. Despite no statistically significant effects at the group level, the linear mixed model analyses revealed substantial individual variability, with random effects (individual trajectories) accounting for marked increases (>50%) in explained variance for several variables (e.g., WEMWBS and general stress). This suggests that while group-level changes were not significant, B-GOS supplementation may have meaningful effects on certain individuals, thereby warranting further exploration. It is also noteworthy that most participants reported very low baseline GI distress and generally high wellbeing, which may have limited the scope for observable improvement. Overall, the ingestion of B-GOS does not lead to enhanced general wellness or exercise-induced GI symptoms, although the high individual variability warrants further research.

In the present study, general stress, mental wellbeing, or sleep was not improved significantly at the group level. This corroborates the findings of other studies reporting no effects on self-perceived stress and anxiety, such as that by Schmidt et al. [5] but also, more recently, that by Tjoelker et al. [26], who reported that consumption of an intervention containing GOS and 2′-FL had no effect on self-perceived stress, anxiety, depression (DASS-42 sub-scores), sleep quality (ASQ), or mood in a group of healthy but stressed Dutch women. This contrasts with one of the most comprehensive studies to date, the “gut feelings” trial [27], which reported a reduction in total mood disturbances (assessed via Profile of Mood States Adult Short Form, second edition) of ~70% with a high prebiotic diet compared with a reduction of only ~10% with the placebo (matched groups: *n* = 28; *n* = 27). Given the lack of randomised controlled trials in this area, the current study adds to the limited literature, especially in recreationally trained endurance runners. On one hand, the findings of this study contest the acting mechanisms of prebiotics (primarily modulating the microbiota–gut–brain axis and reducing chronic inflammation) to improve aspects of performance and wellbeing. However, it may be that the duration of supplementation was not sufficient to see significant changes in performance or wellbeing. Future research with a longer supplementation period is needed to ascertain if benefits could be realised, which may include monitoring symptoms and wellbeing beyond the supplementation period.

Interestingly, in the current trial, substantial individual variability was observed for WEMWBS, with random effects accounting for 73% of the variance. This indicates considerable between-participant differences in response to B-GOS, which may help explain the lack of statistically significant group-level effects. Notably, six participants receiving B-GOS showed an improvement of at least a 3 points in the WEMWBS total score, compared with only two participants in the Placebo group. Given that the 3-point threshold is used for determining clinical meaningful outcomes [28], this suggests that some may benefit from B-GOS supplementation. Accordingly, based on these findings, we speculate that if this study had a longer supplementation period or a larger sample size, it may have led to larger improvements in WEMWBS (given that it took 12 weeks to see any increases versus placebo). Based on the evidence, it is promising yet inconclusive that prebiotics such as B-GOS may improve general wellness. Future studies should continue to test this hypothesis but attempt to standardise methods as best as possible with larger sample sizes.

Exercise-induced GI symptoms did not improve following B-GOS supplementation. Specifically, there was a reduction in upper and lower GI distress by approximately 50% between 0 and 12 weeks for both Placebo and B-GOS. This contrasts with previous research reporting that ingestion of B-GOS for 1 to 24 weeks reduced bloating and abdominal pain [29], traveller’s diarrhoea [30], and GI distress across a season in elite rugby union players [4]. The reason for this discrepancy in the current study may be attributed to the sample (recreational runners), who had low levels of GI distress at baseline and were not undertaking the levels of training of the elite rugby union players who participated in the study by Parker et al. [4]. In the studies reporting a benefit to GI-related distress, all either had previous history of GI distress or were subject to higher levels of stress either acutely or chronically (i.e., across a season). Based on this line of evidence, it is plausible that B-GOS may be suited to individuals who already report high GI and/or training stress, which was not the case in the current study. This is similar to health-related outcomes, where B-GOS has been effective in reducing GI distress for individuals experiencing GI distress [29]. It can be concluded, therefore, that B-GOS supplementation has a limited role in reducing GI distress in recreational endurance runners.

This study also reports no effect of B-GOS supplementation on sport-specific recovery scores. Given the relatively high frequency of vigorous and moderate activity, it was theorised that B-GOS may have elicited an impact on recovery in this population. However, this is perhaps unsurprising given that direct measurements of GI distress, general stress, sleep, and wellbeing were also not statistically significantly different between B-GOS and Placebo in the current study. These findings conflict in part with the preliminary theories that prebiotic ingestion can lower systemic inflammation and subsequently improve recovery [31]. Parker et al. [4] reported that ingestion of B-GOS for 24 weeks improved self-reported upper respiratory symptoms, GI symptoms, and markers of immunity in 33 elite rugby union players, and whilst recovery was not directly measured, players did have greater training availability through reduced illness. Compared with our cohort, it may be that the higher training status and thus levels of exercise stress in the cohort used by Parker et al. [4] may have allowed for the effects of B-GOS to be realised. The use of B-GOS may, therefore, be more suited to highly trained athletes or those completing intense training.

A limitation of this study is that we could not confirm an increase in Bifidobacterium levels, as we did not conduct assessment of the microbiome. However, other studies have repeatedly reported an increase in Bifidobacterium levels following as few as 7 days of supplementation with B-GOS [29,32]. We also acknowledge that the time frame of supplementation was longer than most studies, at 12 weeks, and nutrition intake was not strictly monitored. We cannot, therefore, guarantee no interference from training adaptations or different nutritional behavioural patterns influencing outcomes. However, sleep and activity levels were consistently measured throughout the study, whilst nutrition intake was monitored periodically with strong encouragement for this to remain the same; therefore, this time frame may not have been an issue. Although LMM did not identify statistically significant fixed effects, it revealed substantial random effects, highlighting notable individual variability in responses to B-GOS supplementation. Importantly, the LMM approach allowed us to model individual trajectories across time, capturing participant-specific patterns of change that traditional analysis of (co)variance (ANCOVA) cannot accommodate, thereby offering a more detailed and realistic picture of response variability. The sample size in our study was modest, and no formal a priori power calculation was undertaken given its pilot design. As such, the study was not powered to detect small to moderate effects, and the possibility of Type II error cannot be entirely excluded. Nevertheless, and of importance, these findings provide feasibility and directional insights that will inform the design and sample size calculations of future trials.

## 5. Conclusions

This study investigated the effects of 12 weeks of B-GOS supplementation on general wellness and exercise-induced GI symptoms in recreationally trained endurance athletes. No statistically significant differences were observed between B-GOS and placebo for gastrointestinal or psychological endpoints, although the observed inter-individual variability warrants larger, mechanistically informed trials. Accordingly, future research should focus on targeted populations with low baseline wellness or greater GI distress, where the potential for meaningful improvement is greater, and should employ sufficiently large samples to capture and confirm these individual-level responses.

## Figures and Tables

**Figure 1 nutrients-17-03390-f001:**
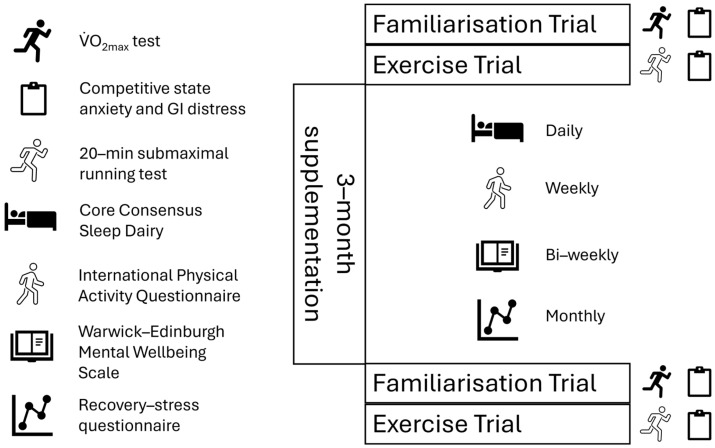
A schematic representation of the study design.

**Table 1 nutrients-17-03390-t001:** Overview of statistical analysis (i.e., linear mixed modelling (LMM)) for each variable in the study following Bimuno (B-GOS) or placebo (PLA) supplementation. * Statistically significant (*p* < 0.05); data in bold denotes Explanatory Power of Random Effects that requires further exploration.

Variable	Group	Time 1 (Mean ± SD)	Time 2 (Mean ± SD)	Change% (Cohen’s D)	Main Effect of Group (β, *p*-Value, CI)	Main Effect of Time (β, *p*-Value, CI)	Interaction (Group × Time) (β, *p*-Value, CI)	Marginal R^2^	Conditional R^2^	Explanatory Power of Random Effects (Change in r2)
**Cognitive A state**	B-GOS	32 ± 3	14 ± 5	−55 (−4.56)	−0.11, *p* = 0.949, [−3.57, 3.34]	**−17.56, *p* < 0.001, [−20.37, −14.75] ***	0.11, *p* = 0.954, [−3.66, 3.89]	0.857	0.912	0.055 (6%)
	PLA	32 ± 3	14 ± 4	−55 (−4.53)						
**Somatic A state**	B-GOS	12 ± 3	13 ± 5	3.6 (0.14)	−0.67, *p* = 0.647, [−3.56, 2.22]	0.44, *p* = 0.639, [−1.47, 2.35]	−0.44, *p* = 0.739, [−3.06, 2.17]	0.780	0.854	0.074 (7%)
	PLA	12 ± 3	12 ± 2	0 (0)						
**State self confidence**	B-GOS	29 ± 3	28 ± 5	−0.8 (0.05)	−2.44, *p* = 0.230, [−6.46, 1.57]	−0.22, *p* = 0.881, [−3.11, 2.67]	−0.44, *p* = 0.832, [−4.25, 3.36]	0.842	0.876	0.034 (3%)
	PLA	26 ± 4	26 ± 6	−3 (0.15)						
**Overall gut discomfort**	B-GOS	1 ± 3	0.1 ± 0.3	−90 (0.74)	−1.11, *p* = 0.073, [−2.30, 0.07]	−1.00, *p* = 0.127, [−2.32, 0.32]	1.11, *p* = 0.227, [−0.67, 2.88]	-	-	-
	PLA	0.1 ± 0.4	0.1 ± 0.3	0 (0.08)						
**Upper GISs**	B-GOS	5 ± 10	3 ± 6	−40 (0.31)	−3.44, *p* = 0.207, [−8.92, 2.04]	−1.89, *p* = 0.456, [−6.19, 2.41]	1.78, *p* = 0.618, [−5.47, 9.03]	0.062	0.199	0.137 (14%)
	PLA	1 ± 2	1 ± 2	−8 (0.02)						
**Lower GISs**	B-GOS	6 ± 12	2 ± 5	−66 (0.62)	−3.22, *p* = 0.285, [−9.24, 2.80]	−4.11, *p* = 0.129, [−9.33, 1.10]	1.67, *p* = 0.654, [−5.90, 9.24]	0.1	0.316	0.216 (22%)
	PLA	3 ± 4	2 ± 1	−81 (0.37)						
**Other GISs**	B-GOS	2 ± 4	2 ± 3	−18 (0.11)	−0.56, *p* = 0.691, [−3.46, 2.34]	−0.33, *p* = 0.774, [−2.50, 1.84]	0.00, *p* = 1.000, [−3.23, 3.23]	0.012	0.327	0.315 (32%)
	PLA	1 ± 2	1 ± 2	−25 (0.11)						
**q3 SLEEP**	B-GOS	17 ± 10	12 ± 8	−29 (0.58)	−3.57, *p* = 0.400, [−12.86, 5.72]	−5.02, *p* = 0.072, [−10.45, 0.42]	0.04, *p* = 0.992, [−7.00, 7.08]	0.131	0.683	**0.552 (56%)**
	PLA	14 ± 9	9 ± 8	−36 (0.57)						
**q4 SLEEP**	B-GOS	2 ± 3	1 ± 2	−37 (0.42)	−1.07, *p* = 0.257, [−2.97, 0.84]	−0.80, *p* = 0.093, [−1.72, 0.11]	1.17, *p* = 0.072, [−0.05, 2.39]	0.048	0.792	**0.744 (74%)**
	PLA	1 ± 1	1 ± 2	35 (0.19)						
**q8 SLEEP**	B-GOS	427 ± 67	371 ± 170	−13 (0.59)	43.65, *p* = 0.346, [−48.45, 135.76]	−55.65, *p* = 0.101, [−121.79, 10.48]	44.90, *p* = 0.320, [−46.34, 136.13]	0.164	0.614	0.450 (45%)
	PLA	471 ± 50	460 ± 45	−2 (0.14)						
**q9 SLEEP**	B-GOS	3 ± 1	3 ± 1	−14 (0.56)	0.54, *p* = 0.192, [−0.28, 1.36]	−0.47, *p* = 0.192, [−1.17, 0.23]	0.52, *p* = 0.285, [−0.49, 1.54]	0.241	0.477	0.236 (24%)
	PLA	4 ± 1	4 ± 1	1.4 (0.06)						
**q10 SLEEP**	B-GOS	3 ± 1	3 ± 1	−7 (0.22)	0.59, *p* = 0.198, [−0.30, 1.48]	−0.20, *p* = 0.470, [−0.79, 0.39]	0.11, *p* = 0.768, [−0.65, 0.88]	0.134	0.694	**0.560 (56%)**
	PLA	3 ± 1	3 ± 1	−3 (0.10)						
**q12a SLEEP**	B-GOS	0.6 ± 0.8	0.2 ± 0.4	−63 (0.63)	−0.14, *p* = 0.635, [−0.72, 0.44]	−0.38, *p* = 0.068, [−0.77, 0.01]	0.52, *p* = 0.065, [−0.02, 1.06]	0.072	0.613	**0.541 (54%)**
	PLA	0.5 ± 0.4	0.6 ± 0.7	31 (0.24)						
**q13a SLEEP**	B-GOS	3 ± 2	3 ± 2	−4 (0.05)	−0.39, *p* = 0.515, [−1.72, 0.95]	−0.79, *p* = 0.134, [−1.73, 0.15]	0.79, *p* = 0.195, [−0.38, 1.97]	0.063	0.774	**0.711 (71%)**
	PLA	3 ± 2	4 ± 3	40 (0.49)						
**q1 IPAQ**	B-GOS	3 ± 2	3 ± 3	−5 (0.07)	−0.33, *p* = 0.789, [−2.85, 2.20]	−0.17, *p* = 0.715, [−1.10, 0.77]	0.67, *p* = 0.312, [−0.64, 1.98]	0.008	0.867	**0.859 (86%)**
	PLA	3 ± 2	3 ± 2	18 (0.22)						
**q2 IPAQ**	B-GOS	40 ± 24	42 ± 24	5 (0.05)	−7.17, *p* = 0.751, [−59.77, 45.42]	2.17, *p* = 0.897, [−30.42, 34.76]	34.50, *p* = 0.163, [−16.42, 85.43]	0.12	0.522	0.402 (40%)
	PLA	33 ± 27	70 ± 72	112 (0.87)						
**q3 IPAQ**	B-GOS	2 ± 2	2 ± 2	22 (0.17)	0.50, *p* = 0.638, [−1.61, 2.61]	0.33, *p* = 0.738, [−1.77, 2.43]	−0.50, *p* = 0.723, [−3.48, 2.48]	0.010	0.148	0.138 (14%)
	PLA	2 ± 2	2 ± 2	−8 (0.08)						
**q4 IPAQ**	B-GOS	15 ± 23	39 ± 52	155 (0.72)	18.00, *p* = 0.316, [−18.58, 54.58]	23.83, *p* = 0.188, [−12.56, 60.23]	−22.17, *p* = 0.381, [−71.99, 27.65]	0.086	-	-
	PLA	33 ± 20	35 ± 29	5 (0.05)						
**q5 IPAQ**	B-GOS	5 ± 2	5 ± 2	0 (0)	1.00, *p* = 0.344, [−1.09, 3.09]	−4.71 × 10^−15^, *p* = 1.000, [−2.05, 2.05]	−1.33, *p* = 0.372, [−4.36, 1.69]	0.075	-	-
	PLA	6 ± 2	5 ± 2	−22 (0.68)						
**q6 IPAQ**	B-GOS	90 ± 84	45 ± 32	−50 (0.20)	20.00, *p* = 0.867, [−234.25, 274.25]	−45.00, *p* = 0.565, [−211.75, 121.75]	168.33, *p* = 0.144, [−57.09, 393.75]	0.109	0.629	**0.520 (52%)**
	PLA	110 ± 138	233 ± 416	112 (0.55)						
**q7 IPAQ**	B-GOS	280 ± 112	320 ± 124	14 (0.29)	75.00, *p* = 0.322, [−107.56, 257.56]	40.00, *p* = 0.392, [−56.78, 136.78]	−60.00, *p* = 0.365, [−195.64, 75.64]	0.046	0.641	**0.595 (60%)**
	PLA	355 ± 164	335 ± 152	−6 (0.13)						
**WEMWBS total score**	B-GOS	51 ± 10	53 ± 7	4 (0.33)	0.25, *p* = 0.941, [−6.78, 7.28]	2.17, *p* = 0.242, [−1.67, 6.00]	−0.42, *p* = 0.862, [−5.47, 4.63]	0.025	0.756	**0.731 (73%)**
	PLA	51 ± 4	53 ± 6	3 (0.26)						
**General stress score**	B-GOS	5 ± 1	5 ± 1	3 (0.08)	0.67, *p* = 0.447, [−1.14, 2.49]	0.14, *p* = 0.796, [−0.99, 1.27]	−0.60, *p* = 0.406, [−2.14, 0.94]	0.026	0.68	**0.654 (65%)**
	PLA	6 ± 2	5 ± 3	−8 (0.27)						
**Recovery-related score**	B-GOS	13 ± 3	14 ± 4	9 (0.35)	1.10, *p* = 0.530, [−2.48, 4.68]	1.20, *p* = 0.278, [−1.13, 3.53]	−1.58, *p* = 0.282, [−4.64, 1.49]	0.020	0.672	**0.652 (65%)**
	PLA	14 ± 3	14 ± 4	−3 (0.11)						
**Sport-specific score**	B-GOS	2 ± 1	3 ± 1	26 (0.36)	1.99, *p* = 0.026, [0.25, 3.72] *	0.61, *p* = 0.505, [−1.25, 2.47]	−0.07, *p* = 0.954, [−2.53, 2.39]	0.300	-	-
	PLA	4 ± 1	5 ± 2	12 (0.32)						
**Sport-specific recovery score**	B-GOS	11 ± 2	13 ± 4	11 (0.36)	0.58, *p* = 0.751, [−3.26, 4.42]	1.29, *p* = 0.262, [−1.11, 3.70]	−1.54, *p* = 0.309, [−4.72, 1.64]	0.018	0.682	**0.664 (66%)**
	PLA	12 ± 3	12 ± 5	−2 (0.07)						

GISs = gastrointestinal symptoms. IPAQ = International Physical Activity Questionnaire. WEMWBS = Warwick-Edinburgh Mental Wellbeing Scales. CI = Confidence interval.

## Data Availability

The data presented in this study are available upon request from the corresponding author due to confidentiality (collaboration agreements).

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
