# Peer review of "The Effects of 12-Week Prebiotic Supplementation on General Wellness and Exercise-Induced Gastrointestinal Symptoms in Recreationally Trained Endurance Athletes: A Triple-Blind Randomised Controlled Pilot Trial"

_nutrients, 2025, doi:10.3390/nu17213390_

Round 1

Reviewer 1 Report

Comments and Suggestions for Authors

Manuscript ID: Nutrients-3933698

Title: The effects of 12-week prebiotic supplementation on general wellness and exercise-induced gastrointestinal symptoms in recreational trained endurance athletes: a triple blind randomized controlled trial

General Comments

This paper reports on an appropriate and newly developing topic, namely, the psychobiotic activity of prebiotic galactooligosaccharides (B-GOS®) in physically active individuals, through a triple-blind randomized controlled approach. The research question is established and the multidimensional outcome (gut symptoms, psychological status, sleep, and activity levels) integration is commendable. The incorporation of linear mixed-model analyses and R² reporting methodology is state-of-the-art when conducted in small-sized experiments.

However, the actual version has a slight weakness in statistical power, lack of reporting of effect magnitudes, insufficient mechanistic contextualization, and over-extension of text without adequate attention to the main points. The data are mostly used to speculate in the discussion section, while the main methodological components (randomization concealment, compliance verification, missing-data handling, multiplicity control) are underexplored.

The study suggests publishable findings once transparency, conciseness, and analytical rigor are optimized.

Specific Comments

Abstract

  1. The objective sentence needed is more concise—stated directly, “This triple-blind RCT investigated whether 12 weeks of B-GOS® supplementation affects gastrointestinal comfort and psychological well-being in recreational athletes.” (lines 15-17)
  2. No quantitative results. Please report n, main model outputs (β ± SE or d values), significance levels for the primary outcomes. (lines 26-31)

Introduction

  1. Reinforce the rationale with a distinction between microbiota-gut-brain axis modulation through prebiotics versus probiotic or synbiotic evidence (lines 39-50)
  2. From 39-66, the manuscript summarizes some representative studies. It flows better with the introduction weaving the literature together in a brisk overview that reinforces the relevant prior evidence in support of the experiment, instead of detailed explanatory documentation of the procedures and results of each study.
  3. It could improve clarity and cohesiveness in this presentation. Add more recent systematic reviews of current research into psychobiotic interventions in athletes. (lines 67-77)

Materials and Methods

  1. Note triple-blind implementation—the participants, investigators, and analysts; describe who was blinded and how randomization codes were held. (lines 80-89)
  2. Specify inclusion/exclusion criteria further (e.g., use of antibiotics, probiotic intake, recent infections, dietary restrictions). (lines 80-89)
  3. There could be indication as to whether a sample-size calculation was done or briefly, if applicable whether there is a specific rationale for the sample taken (i.e., pilot nature). The interpretability of null findings is somewhat limited in the absence of a priori power estimation.
  4. Please outline the monitoring methods for compliance (e.g., capsule counts, adherence percentage) and how adverse events were recorded and reviewed. This added information would increase transparency and reproducibility.
  5. Regarding the questionnaire results you could provide a brief description of total and sub-scores, validation references, and point out the score directionality (i.e., whether higher score = better or worse). This clarifying information would be to the benefit of readers who could easily pick up on it.
  6. The statistical analysis bit is carefully planned and there is a slight amount of additional transparency that would make it stronger. Maybe the random‐effects structure (e.g., which random intercepts/slopes were specified) and model formula need to be explained a bit more explicitly. A brief description on how singular fits were addressed (if models were re‐fitted with linear models) and how missing data were dealt with (listwise deletion versus imputation) would also be useful. Moreover, an initial justification of the method of multiple comparisons, e.g., whether p values were adjusted or false discovery rate was controlled, would also be useful. Finally, give an interpretation of R² in this context for each model reported (marginal versus conditional).

Results

Results. In the Results part, a shorter narrative summary summarizing essential descriptive statistics (like mean ± SD or median [IQR]) would be better so readers aren’t depending on tables. For those outcomes that are not statistically significant, effect sizes with 95% CIs would still serve to convey the magnitude and precision of estimates. The observed improvement in “Cognitive A-state” may be addressed tentatively to show if this improvement persisted after any subsequent multiple-testing modification. Explicitly stating that no group-by-time interactions were detected for the primary endpoints might also be helpful. Lastly, ensuring that Tables 2–4 use consistent units, clearly distinguish “Estimate (SE)” from “Effect size,” and apply uniform decimal precision would enhance overall clarity and consistency.

Discussion

The Discussion should start with the main finding that B-GOS did not produce significant effects before getting to potential explanations. Also, the speculative tone can be moderated in this paper and explicitly state that the small sample size and notable inter-individual variability as limitations also need attention, including a brief note on how those hinder statistical power and precision. Situating the results in the context of the psychobiotic literature in athletes, on which relatively few RCTs have been published, may be even more relevant in terms of how clear and transparent the null findings offered by this study are. And, last, try to combine a few parts of a couple of overlapping paragraphs to improve the reader experience and the overall flow of the story.

Conclusions

For the Conclusions, you wish to put it differently by: (i) emphasizing the preliminary nature of the results, “Within the limits of a small triple-blind RCT, B-GOS® did not significantly affect gastrointestinal or psychological endpoints; the observed inter-individual variability warrants larger, mechanistically informed trials”; (ii) expanding on a short statement about translation (e.g., noting the feasibility of psychobiotic supplementation within athlete monitoring programs as part of pragmatic protocols involving adherence checks and symptom tracking).

Tables and Supplementary

The supplementary tables are extensive, but there are a couple of slight tweaks that could make it even easier to use. Brief definitions (including abbreviations and units of variables, for example), consistent rounding (two decimals), a brief legend for the model type, and a summary for each table can help you to have a more efficient output.

Reviewer 2 Report

Comments and Suggestions for Authors

This manuscript addresses the effects of 12-week B-GOS (Bimuno®) supplementation on gastrointestinal (GI) distress, wellness, sleep, and psychological variables in recreational endurance athletes. The topic is relevant given the increasing use of prebiotics by physically active individuals. The study is generally well organized and written clearly, but the experimental design and the interpretation of non-significant findings limit its scientific contribution. Overall, the manuscript could be suitable for publication after major revision, provided that the authors strengthen methodological transparency, statistical justification, and temper the claims regarding potential benefits.

Abstract

The abstract summarizes the design and main results but reads as overly detailed while lacking interpretive precision. The concluding statement (“promising, albeit not conclusive, positive effects”) is not supported by the data since no group-level differences reached significance.

Introduction

The introduction provides a broad overview of prebiotics, Bifidobacterium, and the microbiota–brain axis but spends too much space summarizing previous literature and too little articulating a specific research hypothesis. Several cited studies involve elderly or clinical populations, which weakens the justification for extending findings to healthy recreational athletes.

Emphasize why recreational endurance athletes (rather than elite or clinical cohorts) represent a relevant target group.

Methods

The description of procedures is detailed and mostly clear. The triple-blind randomized design is commendable, but several methodological limitations reduce confidence in the findings:

  1. Sample size: The study enrolled only 18 participants (9 per group) without a priori power calculation. Even as a proof-of-concept study, minimal justification for sample size is expected.

  2. Dietary control: Nutritional intake was only self-reported for four days, with no analysis presented. This should be summarized or acknowledged as a limitation.

  3. Randomization and blinding: The process is described briefly but requires explicit clarification of who generated the random sequence, who enrolled participants, and how blinding was maintained.

  4. Outcome measures: The inclusion of multiple questionnaires (WEMWBS, REST-Q, IPAQ, CSAI-2) is appropriate, yet their timing and scoring should be better specified.

  5. Statistics: The linear mixed-model approach is appropriate, but the description could be simplified and focused on fixed vs random effects. The duplication of text in sections 3.2–3.3 and Table 1 appears to be a formatting error.

Results

Results are reported in great detail, but the excessive numerical information obscures the main findings. Nearly all variables showed no significant group, time, or interaction effects. Despite this, the authors discuss “promising” individual responses without robust statistical evidence.

  • Avoid repetition of identical sentences (“no significant differences were found…”).

  • Clarify inconsistencies in participant numbers (12 males + 8 females = 20 vs. 18 stated).

Discussion

The discussion acknowledges the null findings and individual variability, but the tone remains overly optimistic. The interpretation should be limited to what the data actually support. References are adequate but primarily drawn from the same research group, which may introduce bias.

  • Reframe the discussion to emphasize feasibility, individual variability, and the exploratory nature of the work rather than potential efficacy.

  • Compare more explicitly with studies showing null results in similar populations.

Conclusion

The conclusion should be shortened and reformulated to align with the results

Reviewer 3 Report

Comments and Suggestions for Authors

Line 94: what method was used to track participant’s adherence to no diet and physical activity changes throughout the study? When I conduct these types of studies, I have participant’s record dietary and physical activity levels to ensure that no significant changes are made.

Line 97: For the vo2 max test, what specific vo2 max test was administered? Were participants connected to a metabolic cart? If not, what values were used to determine that participants actually reached exhaustion?

Reviewer 4 Report

Comments and Suggestions for Authors

The results of the research complement the knowledge contained in numerous publications on the impact of probiotics on the psychophysical state of people. The goals of the research have been achieved, but the mechanism of action of the probiotic Bimuno is unclear. A small number of participants (female 6 or 8 – line 18), a significant dispersion of results and the predominance of subjective indicators reduce the scientific value of the work.

The study is well planned and executed, the work is well edited and can be published.

Nevertheless, in order to increase its value, I propose minor additions:

 Section 2.2 shall further describe the type of diet, including the amount of protein and carbohydrate intake and the follow-up of patients.

In section 2.3, specify which method was used to determine blood lactate and whether both lactic acid isoenzymes (L and D) were included.

 In section 4 (discussion), the likely beneficial mechanism of action of the prebiotic on the investigated parameters should be discussed in more detail, in particular the flow of the gut microbiome on the production of neuroactive metabolites.

Round 2

Reviewer 2 Report

Comments and Suggestions for Authors

accept